# Inhibition of Sphingosine Kinase 2 Results in PARK2-Mediated Mitophagy and Induces Apoptosis in Multiple Myeloma

Jian Wu [ORCID], Shengjun Fan [ORCID], Daniel Feinberg, Xiaobei Wang, Shaima Jabbar and Yubin Kang *

Division of Hematologic Malignancies and Cellular Therapy, Department of Medicine, Duke University Medical Center, Durham, NC 27710, USA
* Correspondence: yubin.kang@duke.edu; Tel.: +1-(919)-668-2331

**Abstract:** Mitophagy plays an important role in maintaining mitochondrial homeostasis by clearing damaged mitochondria. Sphingosine kinase 2 (SK2), a type of sphingosine kinase, is an important metabolic enzyme involved in generating sphingosine-1-phosphate. Its expression level is elevated in many cancers and is associated with poor clinical outcomes. However, the relationship between SK2 and mitochondrial dysfunction remains unclear. We found that the genetic downregulation of SK2 or treatment with ABC294640, a specific inhibitor of SK2, induced mitophagy and apoptosis in multiple myeloma cell lines. We showed that mitophagy correlates with apoptosis induction and likely occurs through the SET/PP2AC/PARK2 pathway, where inhibiting PP2AC activity may rescue this process. Furthermore, we found that PP2AC and PARK2 form a complex, suggesting that they might regulate mitophagy through protein–protein interactions. Our study demonstrates the important role of SK2 in regulating mitophagy and provides new insights into the mechanism of mitophagy in multiple myeloma.

**Keywords:** multiple myeloma; sphingosine kinase 2; PARK2; mitophagy; apoptosis





## 1. Introduction

Multiple myeloma (MM) is a common plasma cell malignancy accounting for more than 17% of hematological malignancies and 1.8% of all cancers in the United States [1]. MM remains an incurable disease, and nearly all myeloma patients eventually relapse from conventional therapies [2]. Therefore, a better understanding of the cellular and molecular mechanisms underlying the pathogenesis of MM is essential for developing effective strategies to treat this devastating disorder and prevent its progression and relapse.

Mitochondria are the main site of adenosine triphosphate (ATP) synthesis in mammalian cells and are critical for most biochemical and physiological processes such as cell growth, survival, and migration [3]. In the past decade, several studies have shown that mitochondria play crucial roles in regulating metabolism, calcium homeostasis, cellular aging, and neurocognitive functions [4–6]. Mitochondrial dysfunction is associated with many diseases, including diabetes, coronary artery disease, aging, and neurodegeneration. More recently, much attention has shifted to the importance of mitochondria in cancer development and progression, as well as to the association between mitophagy and tumor apoptosis. Mitochondria are directly involved in regulating cell death, including apoptosis [7,8]. B-cell lymphoma-2 (Bcl2) family member proteins interact with mitochondria by binding to voltage-dependent anion channels (VDACs). This binding accelerates channel opening and the release of cytochrome c [9]. Additionally, it regulates cancer progression and therapeutic resistance [10]. Myeloid leukemia cell differentiation protein-1 [MCL-1] [11] and Bcl-xl inhibit apoptosis by antagonizing pro-apoptotic members of the Bcl-2 family located at the outer mitochondrial membrane [12]. Moreover, MCL-1 and Bcl-xl regulate mitochondrial homeostasis and bioenergetics by preserving the integrity of the inner mitochondrial membrane and promoting the assembly of ATP-synthase oligomers in the electron transport chain [7].

Mitophagy, or mitochondrial autophagy, clears damaged mitochondria and determines mitochondrial quality and homeostasis [13,14]. However, the role of mitophagy in tumorigenesis remains unclear. Mitophagy is activated in transformed cells and is beneficial for tumor maintenance and progression [15,16]. However, excessive autophagy can act as a tumor-suppressive mechanism, possibly by initiating cell death [17,18]. The most likely explanation for these divergent findings is that mitophagy plays different roles in cancer pathogenesis depending on the stage of the disease, cell types, oncogenic drivers, and activation signal intensity [19–21].

Despite the important role of mitophagy in development and disease, the molecular mechanisms of mitophagy are not well understood. To date, many studies have identified various molecules that are involved in regulating mitophagy. PINK1 (PTEN-induced putative kinase 1)-PARK2 (parkin RBR E3 ubiquitin protein ligase)-dependent mitophagy is the most well-characterized pathway [22]. PINK1 is stabilized and accumulates in the outer membrane, where it binds and recruits PARK2 [23,24]. PARK2 then ubiquitinates and promotes the degradation of several outer mitochondrial membrane proteins, including the mitochondrial fusion proteins MFN1 (mitofusin 1) and MFN 2 [25,26]. Finally, phagophores target the mitochondria via specific receptors such as LC3B. Sequestered mitochondria are then degraded by fusing with lysosomes [27].

Sphingosine kinase 2 (SK2), an enzyme that catalyzes the formation of bioactive lipid sphingosine 1-phosphate (S1P), has recently been identified as a viable target for therapeutic intervention in MM [28,29]. High SK2 expression is involved in various biological processes, including cell growth, survival, and disease pathogenesis [30,31]. Moreover, SK2 expression levels were increased in bone marrow CD138+ myeloma cells from patients [32–34]. ABC294640, an SK2-selective inhibitor, induces caspase 3-mediated apoptosis and inhibits proliferation in MM cells [34]. The role and molecular mechanisms of SK2 in regulating mitophagy in MM are unknown. In this study, we determined the role of SK2 in the mitophagy of MM. Additionally, the underlying mechanisms were investigated using both pharmacological and genetic approaches.

## 2. Materials and Methods

### 2.1. Cell Lines

The MM cell lines used in this study included MM1.R, MM1.S, NCIH929, and U266. The MM1.R (ATCC CRL-2975) and MM1.S (ATCC CRL-2974) cells were purchased from ATCC (Manassas, VA, USA). The NCIH929 (540-CRL-9068) and U266 (TIB-196) cells were purchased from the Duke Cell Culture Facility (CCF). All cell lines were cultured at 37 °C under 5% $CO_2$ in a RPMI1640 medium supplemented with 1% (*v/v*) penicillin and 10% FBS (Mediatech, Herndon, VA, USA). HEK293 cells were maintained in DMEM supplemented with 10% FBS and a 1:100 antibiotic–antimycotic solution.

### 2.2. Antibodies and Reagents

SK2 antibodies were obtained from Santa Cruz Biotechnology (Cat#: SC-398394). PARK2 antibody (#Proteintech, 14060-1-AP), PP2AC antibody (#Cell Signaling, 2038S), AKT antibody (#Cell Signaling, 9272S), SET antibody (#Abcam, ab97596), SET beta antibody, PTEN antibody (#Cell Signaling, 9552S), p-PTEN antibody (#Cell Signaling, 9557S), c-Myc antibody (#Cell Signaling, 9402), caspase 3 antibody, various caspase 9 antibody, LC3B antibody (#Cell Signaling, 2775S), BCL-2 antibody (#BD Bioscience, 610538), and MCL-1 antibody (#Abcam, ab32087) were purchased from commercial sources as indicated. ABC294640 (an SK2-specific inhibitor) was synthesized by Apogee Biotechnology Corp. The mitophagy inhibitor, bafilomycin, was obtained from Sigma-Aldrich (St. Louis, MO, USA). Okadaic acid (OA), a PP2A inhibitor, was purchased from Calbiochem (Gibbstown, NJ, USA).

## 2.3. Cell Proliferation Assay

For the thiazolyl blue tetrazolium bromide (MTT) cell proliferation assay, myeloma cells were plated in triplicate in 96-well plates at a final volume of 100 μL containing $5 \times 10^4$ cells/well and concentrations of ABC294640. The cells were cultured at 37 °C in a 5% $CO_2$ incubator for various durations as indicated. At the time-points indicated, 20 μL of the combined MTS–PMS solution (5 mg/mL MTT) was added onto each well of the 96-well assay plate and incubated for 3–4 h at 37 °C in a 5% $CO_2$ incubator. The absorbance was measured using a ELISA plate reader at 490 nm.

## 2.4. Annexin V–Dye Apoptosis Assay

The annexin V apoptosis assay was performed according to the manufacturer's instructions (G-Biosciences, Cat#786-1548). Briefly, the cells were washed with $1\times$ annexin V–PBS binding buffer and resuspended in 100 μL of annexin V–dye conjugate/propidium iodide staining solution ($1 \times 10^5$ cells) for 15 min at room temperature in the dark. Then, 400 μL of $1\times$ annexin V–PBS binding buffer was added to the cell suspension. Cells were analyzed on a flow cytometer using 488 nm excitation and 525 nm (FL1 channel) emission for annexin–FITC and 730 nm (FL2 channel) emission for propidium iodide.

## 2.5. Western Blot Analysis

The MM cells were harvested, washed with phosphate buffer saline (PBS), and resuspended in a RIPA lysis buffer (#Thermo Scientic, 89900, Rockford, IL, USA) with a cocktail protein inhibitor (#Thermo Scientific, 1862209) on ice for 15 min. The lysates were centrifuged at >$10,000\times g$ for 15 min to remove cell debris. Total protein was quantified using a Dc protein estimation kit (Bio-Rad) with bovine serum albumin (BSA) as a standard curve. Approximately 20 μg of protein was loaded and subjected to SDS-PAGE. The proteins were transferred onto polyvinylidene fluoride membranes. The membranes were blocked with 5% BSA in Tris-buffered saline containing 0.05% Tween 20. Primary antibodies were incubated with 5% BSA in a buffer overnight at 4 °C with gentle rocking. The membranes were then probed with an HRP-conjugated secondary antibody and developed using the Pierce ECL substrate.

## 2.6. Lentivirus Production and Gene Transduction

The production and gene transduction of lentiviruses encoding LC3B, SK2, and control vectors were performed as previously described [27,34]. PINK-specific shRNA, SK2-specific shRNA, and SK2-overexpressing plasmids were obtained from Addgene. PARK2-specific shRNAs were designed using the Block-iT RNAi Designer tool (Invitrogen) (accession number NC_000006.12) with target sequence CTTAGACTGTTTCCACTTATA. The lentiviruses were produced after the transient transfection of HEK 293T cells with an individual lentiviral vector along with packaging plasmids (VSV-G and psPax2) according to the manufacturer's instructions. Supernatants containing viral particles were collected 48 h after transfection. The cells were transduced with lentiviruses by co-centrifugation at $3000\times g$ for 3 h at 37 °C in the presence of 8 μg/mL polybrene.

## 2.7. Mitochondrial Membrane Potential (Δψm) Analysis

Mitochondrial membrane potential (Δψm) was determined using a JC-1 fluorescent probe kit (Molecular Probe, Eugene, OR, USA) as previously described [15]. Briefly, the MM cells were suspended in 1 mL of a warm medium at a density of approximately $1 \times 10^6$ cells/mL. Ten μL of 200 μM JC-1 was added to the cell suspension, and the cells were incubated at 37 °C in 5% $CO_2$ for 15–30 min. The cells were then washed with 2 mL warm PBS, resuspended in 500 μL PBS, and analyzed on a flow cytometer with 488 nm excitation using an emission filter for the Alexa Fluor 488 dye.

*2.8. Co-Immunoprecipitation (co-IP) Assay*

Co-immunoprecipitation of endogenous PP2AC and PARK2 was performed using the Pierce Immunoprecipitation Crosslink Magnetic IP/Co-IP kit (Thermo Scientific, #88805) according to the manufacturer's instructions. Briefly, cell lysates were prepared in a lysis/wash buffer supplemented with 1× Complete Protease Inhibitor Cocktail (Roche, Mannheim, Germany) and 1× PhosSTOP Phosphatase Inhibitor Cocktail (Roche). Protein concentration was determined using Pierce BCA Protein Assay (Thermo Scientific). Four μg of the anti-PP2AC antibody (Cell Signaling, #2038S) or of IgG (serving as a control) was cross-linked to Protein A/G magnetic beads, incubated with 1 mg of protein lysate, and incubated overnight at 4 °C to prepare the immune complex. Then, 10 μL of the antigen sample/antibody mixture was removed as the input for subsequent western blotting assays, and the remaining sample was mixed with magnetic beads and incubated at room temperature for 2 h. The magnetic beads were collected, washed, and eluted in an elution buffer. Lastly, the supernatant was used for western blotting [35].

*2.9. PP2A Activity Assay*

The cells ($1 \times 10^6$ cells) were treated with ABC294640 (30 μM) or transfected with lenti-SK2-specific shRNA (shSK2) or an SK2 overexpression plasmid for 48 h, and then lysed using a RIPA lysis buffer. PP2A activity was measured using a PP2A Immunoprecipitation Phosphatase Assay kit (17-313, Millipore, Temecula, CA, USA). This kit measured the activity of the C subunit of PP2A. Briefly, the protein lysates were incubated with the PP2A antibody at 4 °C with continuous rotation for 2 h. Following the addition of assay buffers and a malachite green solution, the plate was read at an absorbance of 650 nm using a microplate reader (BioTek Instruments, Winooski, VT, USA). Phosphatase activity was determined using a standard curve. The experiments were repeated at least in triplicate, and phosphatase activity was reported by picomoles of phosphate per minute as the mean ± standard error.

*2.10. Immunofluorescence Confocal Microscopy*

The U266 and MM1.R cells were attached to a glass slide coated with 10 mg/mL fibronectin (Sigma, catalog 341635) for 1 h at 37 °C. The cells were subsequently fixed with 4% formaldehyde in PBS for 15 min at room temperature. After fixation, the slides were blocked with 10% FBS in a cell culture medium and subsequently incubated overnight at 4 °C with PP2AC and PARK2 antibodies. The slides were subsequently washed three times with PBS and stained with an Alexa Fluor 594 goat anti-rabbit antibody (Thermo Fisher Scientific, R37117) for 1 h. After washing them three times with PBS, the slides were stained with DAPI (Cell Signaling, 4083) for 5 min and mounted with an antifade mounting medium (Vector, H-1000). Images were acquired using a confocal laser-scanning microscope (Leica SP5 inverted-confocal microscope). Sequential scanning of different channels was performed at a resolution of 1024 × 1024 pixels. The system was equipped with 63 × 1.1HC PLAPO CS2.

*2.11. Seahorse Assay*

MM1.S cells were treated with ABC294640 for 24 h, harvested, and seeded into an XF cell culture microplate (XF24 Flux Pak; Seahorse Biosciences, North Billerica, MA, USA) at a density of 40,000 cells per well. The cells were treated sequentially with oligomycin (1 μM), FCCP (3 μM), and antimycin (2.5 μM) [36]. The oxygen consumption rate was then recorded using an XF24 extracellular flux analyzer (Seahorse Biosciences).

*2.12. Transmission Electron Microscopy (TEM) Assay*

The MM cells receiving different treatments were resuspended, washed with a HBSS buffer three times at room temperature, and fixed with a TEM fixative (10 mL 20% formaldehyde, 4 mL 25% glutaraldehyde, 5 mL 10× PBS, 0.01% malachite green, and 31 mL distilled water) for at least 2 h at room temperature or overnight at 37 °C. The fixative was removed,

and the samples were washed with PBS, post-fixed with $O_SO_4$ for 1 h, blocked with 1% uranyl acetate, dehydrated in ethanol, and flat-embedded in Araldite 502 (Electron Microscopy Sciences, Hatfield, PA, USA). A 60 nm En face section was cut and stained with uranyl acetate and lead citrate using standard methods. Stained grids were examined using a Philips CM-12 electron microscope (EFI, Hillsboro, OR, USA) [27].

### 2.13. Statistical Analysis

Each experiment was performed in triplicates, and values were presented as mean ± standard error of the mean (SEM). Data were analyzed using a Student's *t*-test. *p* values were designated as follows: * $p < 0.05$; ** $p < 0.01$; *** $p < 0.001$; NS means not statistically significant.

## 3. Results

### 3.1. SK2 Expression Is Upregulated in Abnormal Plasma Cells of Patients with MM, and SK2 Overexpression Is Associated with Poor Survival

Given the accumulating evidence relating high SK2 expression to oncogenesis in other types of cancers, we first investigated the expression level of SK2 in patients with MM and the correlation between SK2 expression and clinical outcomes. Using Gene Expression Omnibus (GEO) datasets and the Genomic Scape database (http://www.genomicscape. com/, accessed on 1 June 2020), we observed a significant increase in the expression of SK2 in MM. This increase was observed in dataset GSE6477 when comparing relapsed MM or newly diagnosed MM to normal donors (Figure 1A). We also analyzed SK2 expression from the GSE13591 dataset and found that SK2 was overexpressed in MM (GSE13591, MM vs. ND $p = 0.0384$) (Figure 1B). These data were consistent with our previous data showing the upregulation of SK2, but not of SK1, in primary myeloma cells [34].

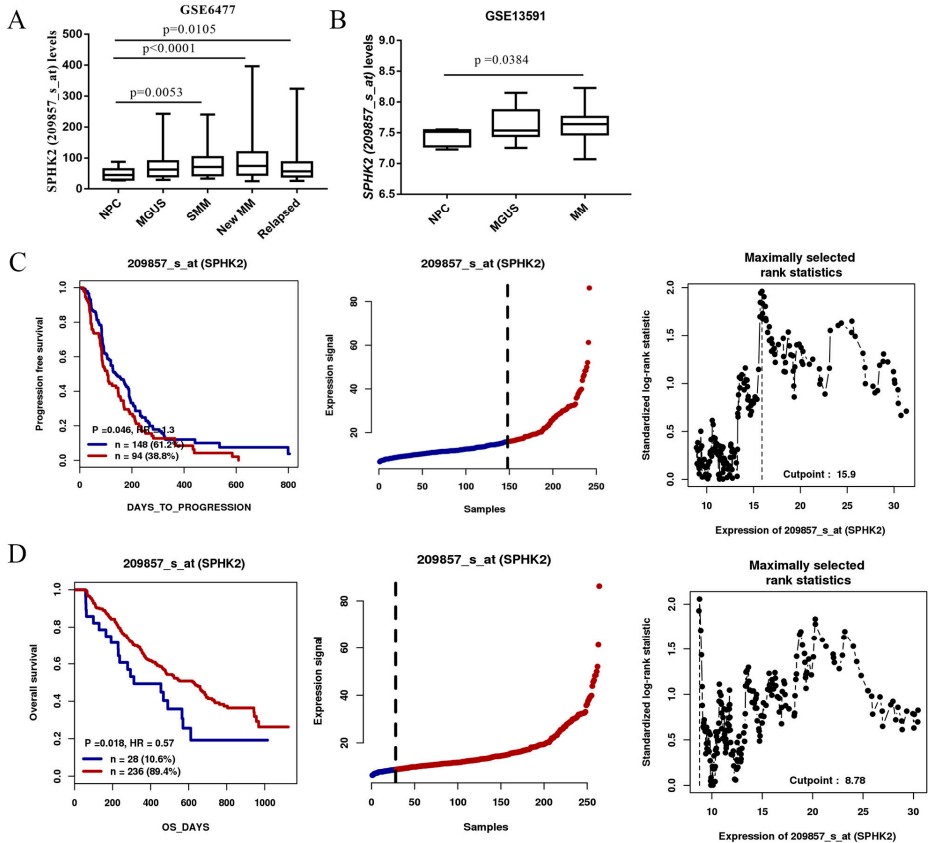

**Figure 1.** SPHK2 elevated in a subpopulation of MM patients and correlation of its overexpression with disease progression and poor survival. (**A**) SPHK2 expression in different stages of MM as

shown in graph; GSE6477 dataset (NPC *n* = 15, MGUS *n* = 22, SMM *n* = 24, newly diagnosed MM *n* = 73 and relapsed MM *n* = 28). (**B**) SPHK2 expression in primary MM cells from GSE13591 dataset (NPC *n* = 5, MGUS *n* = 11, MM *n* = 133). (**C**,**D**) Kaplan–Meier plots indicate progression-free survival (PFS) and overall survival (OS) of 528 MM patients in APEX clinical trials (GSE9782) categorized by SPHK2 expression. *p*-value-determined log-rank test. NPC: normal plasma cells; MGUS: monoclonal gammopathy of undetermined significance; SMM: smoldering multiple myeloma.

To investigate the prognostic significance of SK2 overexpression in MM development and progression, we evaluated SK2 gene expression in the APEX trial GEO microarray database (GSE9782) and correlated it with clinical outcomes such as progression-free survival (PFS) and overall survival (OS). High SK2 expression correlated with significantly shorter PFS (106 days vs. 140 days, *p* = 0.028) and OS (312 days vs. 628 days, *p* = 0.0011). Consistent with our previous findings, these data strongly suggest that SK2 plays a critical role in MM pathogenesis.

### 3.2. SK2 Inhibition Induces Mitophagy in MM Cells

SK has two isoforms, SK1 and SK2. SK1 is mainly cytosolic, whereas SK2 is predominantly localized in the mitochondria, endoplasmic reticulum, and nucleus [37–39]. Recent studies from several laboratories, including ours, have demonstrated that PINK1-PARK2-dependent mitophagy plays an important role in the carcinogenesis of MM [27] and pancreatic cancer [40]. Therefore, we aimed to determine whether SK2 regulates mitophagy in MM cells.

A loss of mitochondrial membrane potential (MMP) is a hallmark of mitophagy, representing an early event in mitochondrial damage coinciding with caspase activation. First, we determined the effect of SK2 inhibition on mitochondrial membrane depolarization. We treated the U266 and MM1.R cells with ABC294640 or genetically knocked down SK2 using shRNA and then measured mitochondrial membrane depolarization using JC-1 MitoProbe. With intact mitochondrial membrane potential, JC-1 remained a monomer (measured in the right upper quadrant, Figure 2A). However, upon mitochondrial membrane depolarization, JC-1 forms dimers and accumulates as aggregates (measured in the right lower quadrant, Figure 2A). Treatment with ABC294640 or genetically inhibiting SK2 induced mitochondrial membrane depolarization, as evidenced by the increased percentage of JC-1 aggregates in the right lower quadrant of the flow cytometer (Figures 2A and S2).

Next, we used the Seahorse XF assay to measure and quantify the rate of ATP production and the mitochondrial system in the MM cells treated with ABC294640. ABC294640 decreased the oxygen consumption rate and extracellular acidification rate in a dose-dependent manner, and significantly reduced ATP production and spare respiratory capacity (Figure 2B), consistent with the mitophagy induction.

Mitophagy is characterized by the fusion of mitochondria and lysosomes. Thus, the MM cells were treated with ABC294640, and transmission electron microscopy was performed to look for mitochondrial and lysosomal fusion. The ABC294640 treatment induced mitophagy, as shown by the large increase in the fusion of the mitochondria with the lysosomes (Figure 2C). Additionally, we monitored the interaction between the mitochondria and the lysosomes using Mito Tracker Red with Lysotracker Green staining, which tracks the colocalization of mitochondria and lysosomes in cells. After ABC294640 treatment, the lysosomal staining intensity and the number of lysosomes colocalized with mitochondria were strongly increased in the U266 and MM1.R cells (Figure 2D).

Additionally, the protein expression levels of key molecules in mitophagy (PINK1, PARK2, and LC3B) were analyzed using a western blot (Figure 2E). PINK1, PARK2, and LC3B expression was upregulated, whereas SK2 levels were decreased in ABC294640-treated or lenti-shSK2 transduced MM cells. These data indicate that inhibiting SK2 expression induced MMP loss, mitochondria–lysosome fusion, and mitophagy.

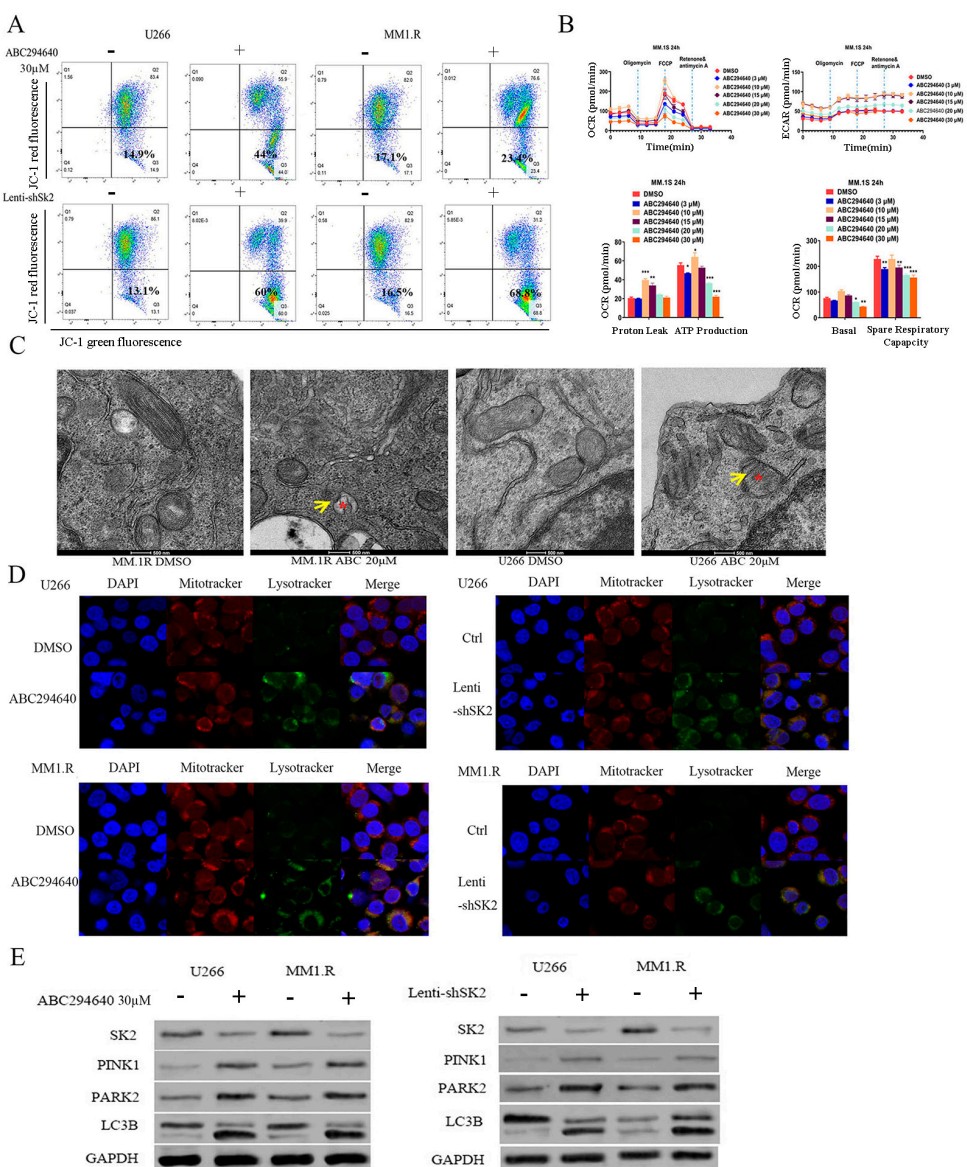

**Figure 2.** ABC294640 inhibition of SK2, and SPHK2 knockdown-induced mitochondrial damage and mitophagy in MM cells. (**A**) Flow cytometric analysis of MM1.R and U266 cell lines using Mito Probe JC-1 Assay Kit. (**B**) O$_2$ consumption rate (OCR) measured in real time under basal conditions and in response to indicated mitochondrial inhibitors. Spare respiratory capacity (SRC) is shown at top. *: $p < 0.05$; **: $p < 0.01$; ***: $p < 0.001$. (**C**) Morphological ultrastructural appearance of mitochondria observed by TEM in U266 and MM1.R cell lines. Asterisks indicate the fusion of mitochondria and lysosome. Yellow arrows indicate enlarged views of mitochondria. (**D**) Inhibited SK2 expression through ABC294640 50 μM or lenti-shSK2 for 48 h; MM1.R and U266 cell lines were co-stained with LysoTracker and MitoTracker and observed by confocal microscopy. (**E**) MM1.R and U266 cells treated with ABC294640 or lenti-shSK2 for 48 h; protein lysate was subjected to western blot with indicated antibodies. Data are representative of at least two independent experiments.

### 3.3. Mitophagy Plays a Crucial Role in Mediating ABC294640 Induced Apoptosis in MM Cells

We previously found that ABC294640 induces apoptosis and inhibits myeloma cell growth both in vitro and in vivo. To determine if mitophagy activation mediated the apoptotic effects of ABC294640, we used bafilomycin, a mitophagy inhibitor, to block mitophagy. Bafilomycin (0.5 nM) was added to the U266 and MM1.R cells treated with ABC294640 for 48 h (Figure 3A). Treatment with bafilomycin reversed the inhibitory effect of cell viability mediated by SK2 inhibition. Moreover, the MM cells exhibited a marked decrease in

apoptosis when ABC294640 was combined with bafilomycin (Figure 3B). Apoptosis markers caspase 3, caspase 9, Bcl-2, Mcl-1, and c-Myc were also measured in ABC294640-treated U266 and MM1.R cells with or without bafilomycin (Figure 3C). We observed that SK2 inhibition via ABC294640 or lenti-shSK2 increased the expression of cleaved caspase 3 and caspase 9, and downregulated the expression of Bcl-2, Mcl-1, and c-Myc; however, SK2 inhibition was reversed by bafilomycin co-treatment. Taken together, these data suggest that blocking mitophagy through bafilomycin reverses the apoptotic cell death induced by SK2 inhibition. This shows that mitophagy induction mediates the apoptotic pathways caused by SK2 inhibition.

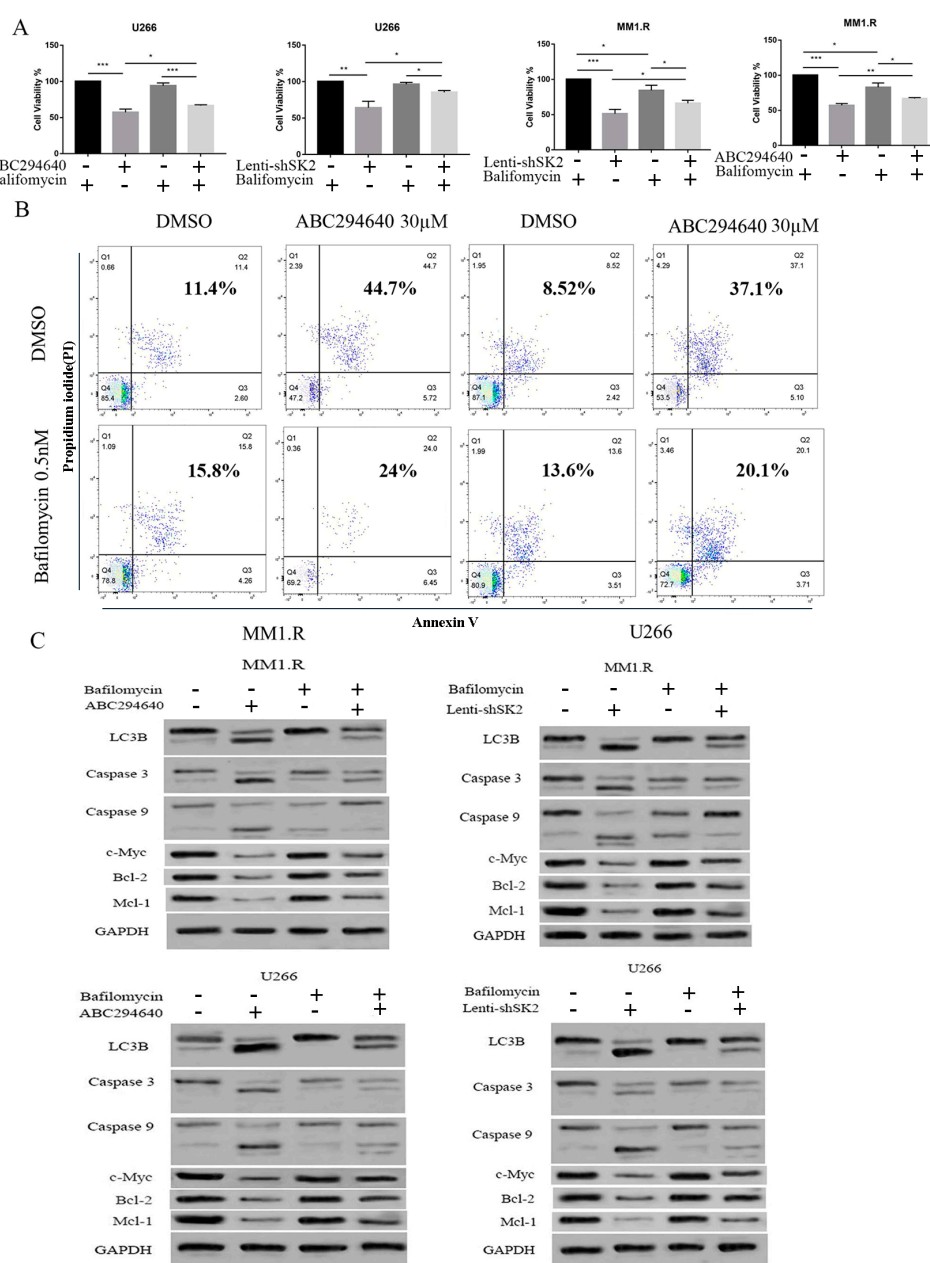

**Figure 3.** Bafilomycin-blocked autophagy blunting effects of ABC294640. SK2 of MM1.R and U266 MM cells inhibited via ABC294640 30 μM or lenti-shSK2 and co-treated with or without 0.5 nM bafilomycin. (**A**)MM cell viability evaluated by MTT assay. (**B**) Co-treated MM cells evaluated by flow cytometry for viability with annexin V. (**C**) Co-treated MM1.R and U266 cells examined for apoptosis-related proteins by western blot. *: $p < 0.05$; **: $p < 0.01$; ***: $p < 0.001$; error bars presented as SD of mean of three separate experiments.

### 3.4. SK2 Inhibition Promotes Crosstalk between PP2AC and PARK2 Mediated by AKT/c-Myc/SET Signaling

Next, we determined the molecular pathways of SK2-regulated mitophagy. The serine-threonine protein kinase AKT1 is an oncogene that selectively regulates PINK1-dependent mitophagy [41,42]. PTEN, a tumor suppressor [43] and dual functional phosphatase [44], is considered an important negative regulator of the PI3K/Akt pathway [45,46]. PTEN inhibits PINK1-PARK2-mediated mitophagy [47,48]. To determine the effects of SK2 on the expression levels of Akt and PTEN, we treated the MM cells with ABC294640 or genetically knocked down SK2 using shRNA (Figures 4A and S4). Additionally, SK2 was overexpressed in the MM cells (Figures 4B and S4). The pharmacological inhibition of SK2 or the genetic downregulation of SK2 resulted in a decrease in Akt expression and an increase in PTEN levels. In contrast, SK2 overexpression increased Akt expression and decreased PTEN expression.

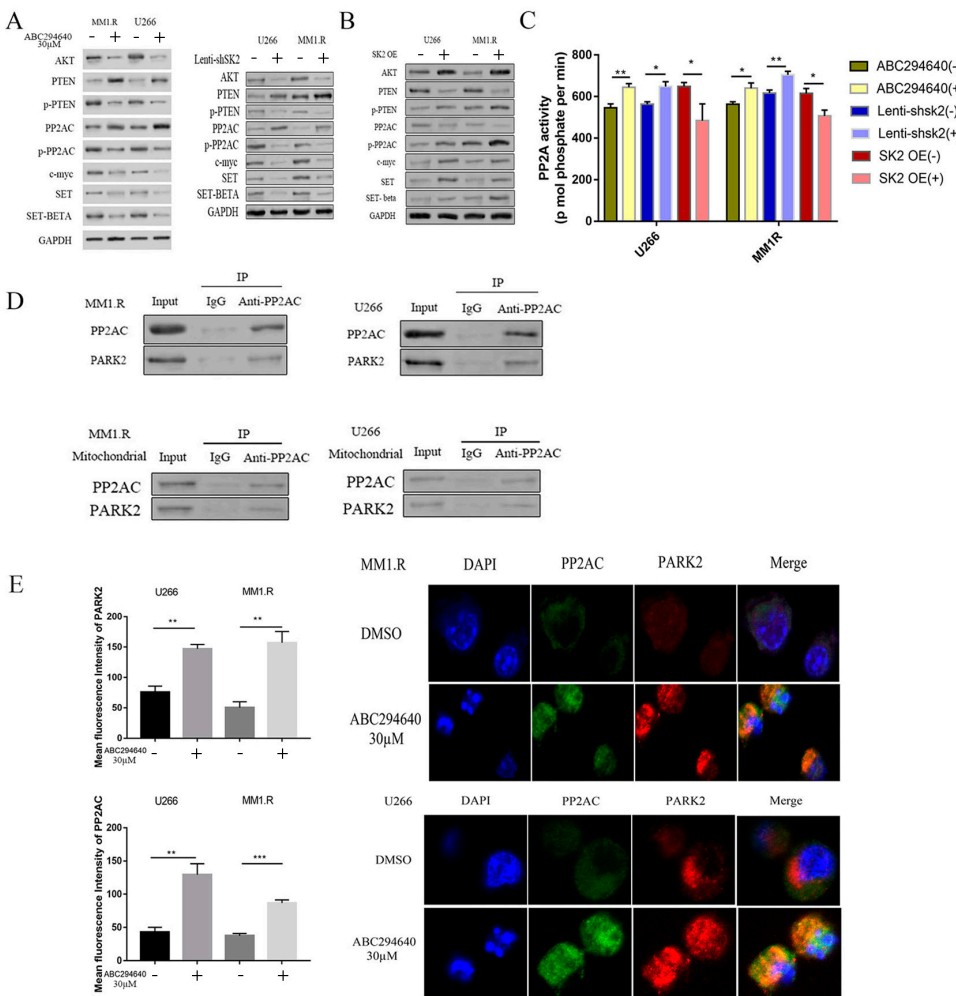

**Figure 4.** Promotion of crosstalk between PP2AC and PARK2 mediated by AKT/c-Myc/SET signaling by inhibition of SK2. (**A**) SK2 inhibition by ABC294640 or lenti-shSK2 in U266 and MM1.R cell lines performed for 48 h; protein lysate harvested and western blots performed with antibodies. (**B**) U266 and MM1.R transduced with an SK2 overexpression plasmid or a control plasmid, and protein harvested at 48 h; harvested protein lysates were examined via western blotting using antibodies as indicated. (**C**) PP2A activity assessed by an activity assay that detected the phosphatase activity of PP2A. (**D**) Co-immunoprecipitation (co-IP) of endogenous PP2AC for PARK2; total protein lysates or mitochondrial protein lysates were immunoprecipitated using an anti-PP2Ac antibody or IgG

control, and pull-down was subsequently probed for PARK2. (**E**) PP2AC (green), PARK2 (red), and DAPI(blue) co-stained in U266 and MM1.R cells to determine localization; slides were imaged with a Leica SP5-inverted confocal microscope. Bar graphs displaying results of mean intensity of immunofluorescence of PP2AC and PARK2. *: $p < 0.05$; **: $p < 0.01$; ***: $p < 0.001$; error bars presented as SD of mean of three separate experiments.

We then determined how SK2 regulates PP2A expression. The SET oncogene is a physiological inhibitor of PP2A, and there are two alternatively spliced SET isoforms: SET alpha and SET beta [49]. SET oncoproteins participate in cancer progression by affecting multiple cellular processes, including the control of the cell cycle, gene transcription, apoptosis, cell migration, and epigenetic regulation. SET contributes to tumorigenesis by forming an inhibitory protein complex with PP2A [50]. SET expression further exacerbates the effects of uncontrolled signaling by inhibiting the endogenous regulators of these pathways. We found that ABC294640 treatment or SK2 shRNA knockdown downregulated SET and SET beta expression (Figure 4A).

Our study focused on protein phosphatase 2A (PP2A). PP2A is an important regulator of signal transduction pathways and is a tumor suppressor gene. PP2A negatively regulates multiple pro-growth/pro-survival signaling pathways associated with cancer progression, such as Akt and c-Myc [51,52]; SK2 was previously found to regulate PP2A activity. PP2A can deactivate Akt. The PP2A core enzyme comprises a 65kD scaffold subunit (known as the A or PR65 subunit) and a 36kD catalytic subunit (or C subunit) [53]. PP2AC phosphorylation leads to PP2A inactivation [54–56]. The pharmacologic inhibition of SK2 or the genetic downregulation of SK2 increased PP2A expression and the activation of PP2A (Figure 4A). In contrast, SK2 overexpression downregulated PP2A expression and inhibited PP2A (Figure 4B). We also measured PP2A phosphatase activity and found that SK2 inhibition increased PP2A phosphatase activity (Figure 4C). These data demonstrate the important role of SK2 in regulating PP2A activity.

We determined how PP2A regulates mitophagy. PARK2 plays a key role in mitophagy. We performed a co-immunoprecipitation of PP2A and PARK2, and found that the PP2A and PARK2 formed a complex and interacted with each other (Figure 4D). We further fractionated the mitochondria and cytosol and found that the PP2A interacted with the PARK2 in the mitochondria (Figure 4D). Double immunofluorescence labeling was performed to confirm this finding (Figure 4E). Treatment with ABC294640 enhanced the interaction between the PP2AC and PARK2.

### 3.5. Inhibition of PP2AC Expression Blocks the Effect of ABC294640

We next sought to determine the role of PP2A in ABC294640-mediated effects. Okadaic acid (OA), a specific inhibitor of PP2A phosphatase activity at concentrations lower than 50 nM, was utilized in an amount of 20 nM [57]. The U266 and MM1.R cells were treated with ABC294640 for 48 h with or without OA, and cell viability was assessed. An MTS assay demonstrated that the co-treatment with OA markedly reversed the inhibitory effect of ABC294640 on cell viability (Figure 5A). To assess the combined effects of the ABC294640 and OA, the combination index (CI) value for each dose was calculated using the CompySyn software based on the Chou–Talalay method. The CI value was found to be greater than one, consistent with the antagonistic effect. The co-treatment with OA reduced the cytotoxic effect of ABC294640 at certain concentrations. (Supplementary Figure S5).

Furthermore, this reversal of the inhibitory effect on cell proliferation by OA was associated with a decrease in apoptosis. Blocking PP2AC decreased the number of apoptotic cells induced by the treatment with ABC294640 in the U266 and MM1.R cells (Figure 5B). Lastly, OA reduced the mitophagy induced by the ABC294640 treatment. Confocal immunofluorescence microscopy images of the U266 and MM1.R cells treated with the ABC294640 and/or OA were analyzed after staining the mitochondria with the MitoTracker dye (deep red) (Figure 5C). The treatment with ABC294640 increased the population of damaged-mitochondria-containing cells. The ABC294640-induced mitochondrial damage was then blocked by OA, as evidenced by the increase in Mito Tracker signals in the combination

group. Similarly, the JC-1 results showed a similar trend: PP2AC inhibition by OA decreased mitochondrial membrane depolarization, in contrast to what happened when the MM cells were exposed to ABC294640 (Figure 5D). These data suggest that inhibiting PP2AC plays an important role in SK2-inhibition-induced PARK2-mediated mitophagy.

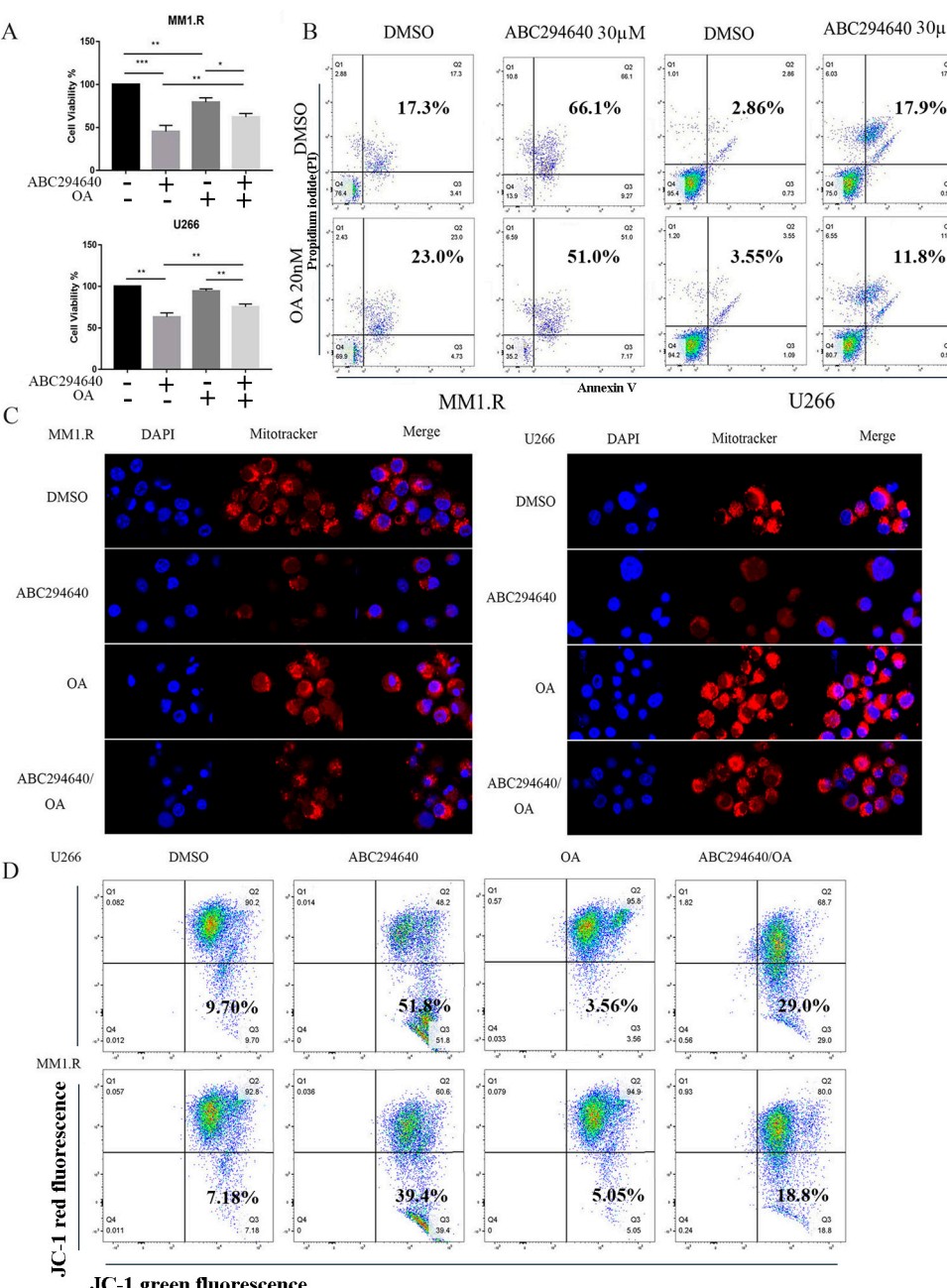

**Figure 5.** Inhibition of PP2AC by okadaic acid (OA)-suppressed mitophagy induction from ABC294640 treatment. Multiple myeloma cells U266 and MM1.R were co-treated with ABC294640 30 μM and/or OA 20 nM for 48 h. (**A**) MM cell viability evaluated by MTT assay. (**B**) Co-treated MM cells analyzed by annexin V/propidium iodide and analyzed by flow cytometry to determine percentage of apoptotic cells. (**C**) MM1.R and U266 stained with MitoTracker and observed by Leica SP5-inverted confocal microscope. (**D**) Flow cytometric analysis of MM cells using Mito Probe JC-1 Assay Kit. *: $p < 0.05$; **: $p < 0.01$; ***: $p < 0.001$; error bars presented as SD of mean of three separate experiments.

### 3.6. Knockdown of PINK1 or PARK2 Expression Attenuates ABC294640-Mediated Mitophagy

To further confirm the role of the PINK1-PARK2 pathway in the ABC294640-mediated mitophagy, we knocked down PINK1 or PARK2 using specific shRNA in myeloma cell lines. The U266 and MM1.R cells lines were treated with 30 µM ABC294640 for 48 h with or without the knockdown of PINK1 or PARK2. The knockdown of PINK1 decreased the level of LC3B and PARK induced by the ABC294640 (Figure 6A, left panel). Similarly, the knockdown of PARK2 alleviated the upregulation of PINK1 and LC3B induced by the ABC294640 treatment (Figure 6A, right panel). Moreover, the JC-1 assay was performed to evaluate the mitochondrial membrane potential. PINK1-specific shRNA knockdown or PARK2-specific shRNA knockdown restored the change that occurred on the mitochondrial membrane, and attenuated the mitochondrial membrane depolarization induced by the ABC294640 (Figure 6B,C). These data demonstrated the important role of the PINK1-PARK2 pathway in ABC294640-induced mitophagy.

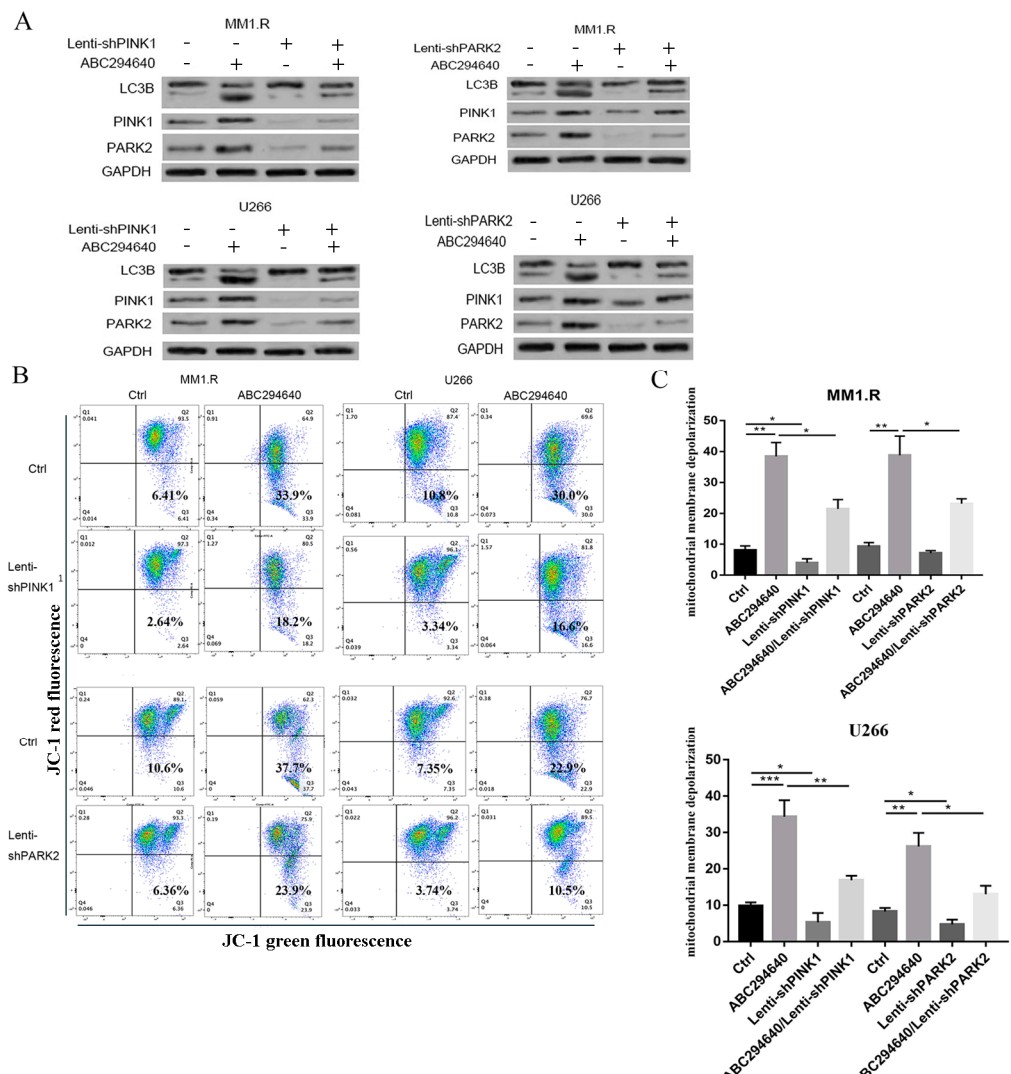

**Figure 6.** ABC294640-mediated mitophagy attenuated by knockdown of PINK1 or PARK2. U266 and MM1.R cells were treated with 30 µM ABC294640 for 48 h with or without PINK1 or PARK2 knockdown. (**A**) Protein lysate was harvested and western blots performed with antibodies. (**B**) Flow cytometric analysis of MM cells using Mito Probe JC-1 Assay Kit. (**C**) Bar graph showing mitochondrial membrane depolarization in cell lines above. *: $p < 0.05$; **: $p < 0.01$; ***: $p < 0.001$; error bars presented as SD of mean of three separate experiments.

## 4. Discussion

Our current study interrogated existing datasets and demonstrated that high SK2 expression occurred in patients with MM and that high SK2 levels were associated with poor clinical outcomes. Our findings are consistent with those of studies on solid tumors, in which the upregulation of SK2 were associated with tumor aggressiveness and poor outcomes [58,59]. Furthermore, the knockdown of SK2 with shSK2 inhibited MM cell proliferation and induced cell death. These data demonstrate the important role of SK2 in MM pathogenesis. Using the SK2-specific inhibitor, ABC294640, we showed that SK2 is a potential therapeutic target of treating MM. Additionally, we investigated the potential mechanism of ABC294640 in MM cell apoptosis. Our previous study showed that ABC294640 induces apoptosis in primary human CD138+ cells and MM cell lines [32]. Herein, we further demonstrate that ABC294640 activates mitophagy through the crosstalk between PP2AC and PARK2 which induces the apoptosis of MM cells. A PINK1- and PARK2-specific knockdown assay confirmed the role of this pathway in ABC294640-induced mitophagy. The inhibition of PP2AC by OA blocked the effects of the ABC294640. Our study provides direct evidence that SK2-mediated mitophagy plays a critical role in regulating myeloma apoptosis, and further elucidates the molecular mechanisms of SK2-mediated mitophagy. We also show that SK2 levels are a potential prognostic biomarker of MM. Taken together, these findings significantly advance our understanding of MM.

ABC294640 is currently undergoing phase I/II clinical trials for myeloma (see trial NCT01410981) [32,60]. ABC294640 induces proteasome degradation and the downregulation of Mcl-1 and c-Myc but has no effect on Bcl-2 expression [32]. The combination of the ABC294640 and a Bcl-2 inhibitor (ABT-199) had a synergistic cytotoxic effect on the MM cells both in vitro and in vivo. Furthermore, ABC294640 showed good pharmacokinetics, oral bioavailability, and bioavailability [30]. Plasma concentrations can reach >200 $\mu$M without hematologic or major organ toxicity, approximately six- to seven-fold higher than the IC50 found in MM cell lines [61]. ABC294640 also had anticancer effects on various solid tumors [62]. ABC294640 causes cell cycle arrest in the S phase and increases the apoptosis rate in epithelial ovarian cancer cells [63]. An in vivo assay also showed the inhibitory effect of ABC294640 on tumor growth [64]. Our current study provides additional justification for testing ABC294640 in clinical settings with relapsed MM patients.

The induction of apoptosis in tumor cells is an important goal in cancer chemotherapy [65]. Mechanistically, we found that ABC294640 induced apoptosis by stimulating mitophagy through the crosstalk between PP2AC and PARK2. Interestingly, the expression levels of phosphatase PP2AC have been reported as having an influence on the interplay between p38$\alpha$ and mTOR signaling [66]. In addition, ABC294640 downregulates pS6 expression [34,58]. This finding suggests that SK2 may also play a role in the mTOR signaling pathway and affect cell translation. Moreover, our previous study demonstrated that c-Myc and Mcl-1 expression levels were reduced by ABC294640 treatment via the proteasome degradation pathway in MM cell lines [32]. These data suggest that c-Myc and Mcl-1 may play major roles in mediating ABC294640-induced apoptosis. To further support this rationale, we performed the JC-1 assay to detect MMP in the MM cells and found that ABC294640 could induce mitochondrial membrane damage and cause early apoptosis. These data indicate that ABC294640 acts via the mitochondria-mediated apoptotic pathway.

The relative roles of SK1 and SK2 in tumor biology have been of great interest to many investigators, and this has been a central issue in the selection of ABC294640 for our study. SK1 is a cytosolic protein that translocates to the cell membrane upon activation and is necessary for tumor progression [67]. While SK1 and SK2 share kinase homology and are 80% similar, SK2 contains distinct, yet unidentified, localization and export signals, which differentiate its cellular localization and biological functions from those of SK1 [68,69]. SK2 also contains a pro-apoptotic BH3 domain that promotes apoptosis when overexpressed [70,71]. However, the exact role of SK2 in the pathogenesis of MM remains unclear. It remains to be determined whether the overexpression of SK2 in MM cells serves as a driving event to initiate the development of MM, or if it merely reflects phenotypic

changes due to other oncogene aberrations. We are currently studying SK2-knockout mice to determine the incidence and severity of myeloma. These studies will help us to further define the role of SK2 in MM development.

PINK1-PARK2-mediated mitophagy can result in either the recruitment and activation of the E3 ubiquitin ligase PARK2 and the downstream autophagy receptor SQSTM1, or in the PARK2-independent recruitment and activation of autophagy receptors like OPTN and CALCOCO2 [72]. PARK2 then localizes to the mitochondrial membrane and is phosphorylated to activate mitophagy [73]. Our data demonstrate the important role of PP2AC in ABC294640-mediated mitophagy and the interaction between PP2AC and PARK2. We found that the inhibition of PP2AC using okadaic acid rescued the mitophagy induced by the ABC294640. Our co-immunoprecipitation procedure clearly demonstrated that the PP2AC interacted with the PARK2 in the mitochondria (Figure 4D). The confocal microscope was less definitive, likely due to the high levels of PP2AC and PARK2 in other subcellular compartments. Despite this, our confocal microscope imaging showed the merge/colocalization of PP2AC and PARK2 (Figure 4E), further evidencing the interaction between PP2AC and PARK2. It is possible that the PP2A regulated the PARK2 phosphorylation level. However, the exact mechanism may be rather complicated, and an investigation of such mechanisms is beyond the scope of this study. In future work, we plan to conduct additional experiments to explore this mechanism in greater depth.

## 5. Conclusions

In summary, our findings show that SK2 is aberrantly upregulated in MM cells and that the inhibition of SK2 through ABC294640 suppresses MM cells. Importantly, the inhibition of SK2 activates mitophagy through the crosstalk between PP2AC and PARK2 to induce apoptosis. Decreased PP2AC expression blocked the effects of the ABC294640. Our study provides important insights into the mechanisms of action of ABC294640 and could lead to the development of novel therapeutic interventions.

**Supplementary Materials:** The following supporting information can be downloaded at: https://www.mdpi.com/article/10.3390/curroncol30030231/s1, Figure S1: SK1 (219257_s_at) expression in different stages of MM and the relationship with overall survival; Figure S2: Inhibition SK2 expression through ABC294640 or lenti-SK2 result to mitochondrial membrane potential changed; Figure S3: Using bafilomycin reduced the ABC294640-induced mitophagy; Figure S4: Inhibition of SK2 involved AKT/c-Myc/SET pathway; Figure S5: The graphic representations obtained from the CompuSyn Report for ABC294640 and bafilomycin in MM1.R and U266; Figure S6: The raw data and intensity ratio of Figure 2E; Figure S7: The raw data and intensity ratio of Figure 3; Figure S8: The raw data and intensity ratio of Figures Figure 4 and S4; Figure S9: The co-IP raw data of Figure 4D; Figure S10: The raw data and intensity ratio of Figure 6.

**Author Contributions:** The study's conception and design were undertaken by J.W., S.F., D.F., and Y.K. Part of the experiment was performed by J.W., S.F., D.F., S.J., and X.W. The statistical analysis and interpretation were performed by J.W., S.F., and D.F. Patient data extraction was conducted by X.W. and Y.K. The original draft was written by both J.W. and Y.K. and edited by D.F. Critical revision and editing were undertaken by all authors. All authors have read and agreed to the published version of the manuscript.

**Funding:** This research was funded by National Cancer Institute to YK. Grant number: R44CA199767, R01CA197792, R21CA234701, R21 CA267275.

**Institutional Review Board Statement:** Not applicable.

**Informed Consent Statement:** Not applicable.

**Data Availability Statement:** The data presented in this study are available on request from the corresponding author.

**Acknowledgments:** The authors thank Emily Chu for proofreading the manuscript. This project was supported in part by the developmental funds of the Duke Cancer Institute as part of the P30 Cancer Center support grant (grant ID: NIH CA014236).

**Conflicts of Interest:** The authors declare no conflict of interest.

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
