# Peer review of "Inhibition of Sphingosine Kinase 2 Results in PARK2-Mediated Mitophagy and Induces Apoptosis in Multiple Myeloma"

_curroncol, doi:10.3390/curroncol30030231_

Round 1

Reviewer 1 Report (Previous Reviewer 1)

The revised version of the manuscript is improved by addressing the questions and considering the suggestions/comments; thus, it is suggested for publication in Current Oncology.

Reviewer 2 Report (Previous Reviewer 2)

The authors demonstrated the inhibition of sphingosine kinase 2 in myeloma by PARK2 mitophagy and apoptosis. The article is well prepared and good for published in the Journal.

This manuscript is a resubmission of an earlier submission. The following is a list of the peer review reports and author responses from that submission.

Round 1

Reviewer 1 Report

This comprehensive study extensively investigated the involvement of SK2 in the regulation of mitophagy and the downstream SET/PP2AC/PARK2 regulatory pathway from a mechanistic perspective.

The study employed a multitude of advanced techniques covering biochemical, genetic, and microscopical approaches.

Involvement of SK2 was previously shown to be involved in the regulation of autophagy. However, this study expands the perspective and dissects the function of SK2 in mitophagy, making the study original and sound.

I have a few minor questions and suggestions that would help the authors improve the manuscript before it is accepted.

1- A rescue condition is suggested for the study to clearly distinguish the proposed mitophagy-driven pathway from the conventional autophagy-driven pathway: For that, it is suggested to include conditions like suppression and overexpression of PINK1/PARK2 mitophagy regulators under SK2 suppression and overexpression conditions and subsequently measuring the changes in the levels of autophagosome markers.

2- There is no clear colocalization profile between PP2AC and PARK2 (Fig.4E); the subcellular fluorescence-intense clusters are predominantly not colocalizing. However, coIP suggests an interaction between these two proteins. Could authors please discuss this discrepancy and what could be the reasons for this observation? Or, if it is not the case, it is suggested to provide a better representative image for a clear colocalization.

3- The image quality in all the figures is low. It seems a document conversion issue. It is suggested to keep them as high quality as possible. Also, some of the font sizes of text labels in the figures are small, making them hard to read.

Reviewer 2 Report

The authors used cell lines to study the SK2 pathway in myeloma with positive results.

1. Since the study was from cell lines, how comes could the fist sentence in discussion section " ...we found the SK2 was highly expressed in MM and...with poor outcome"?

2. How about SK2 status in MM patients?